# Evaluation of Rye Bran Enzymatic Hydrolysate Effect on Gene Expression and Bacteriocinogenic Activity of Lactic Acid Bacteria

**Julia M. Epishkina** [1], **Maria V. Romanova** [1] , **Marina A. Chalenko** [1], **Natalya Yu. Khromova** [1], **Boris A. Karetkin** [1,*] , **Andrey V. Beloded** [1] , **Maria A. Kornienko** [2], **Yulia M. Averina** [3], **Irina V. Shakir** [1] and **Victor I. Panfilov** [1]

[1] Department of Biotechnology, Faculty of Biotechnology and Industrial Ecology, Mendeleev University of Chemical Technology, 125047 Moscow, Russia

[2] Federal Research and Clinical Center of Physical-Chemical Medicine of Federal Medical Biological Agency, 119435 Moscow, Russia

[3] Department of Logistics and Economic Information, Mendeleev University of Chemical Technology, 125047 Moscow, Russia

* Correspondence: karetkin.b.a@muctr.ru; Tel.: +7-495-495-2379

**Abstract:** Lactic acid bacteria (LAB) bacteriocins can be considered as a bio-preservatives and an alternative to antibiotics, but the high manufacturing costs limit their commercial application. The screening of LAB strains for bacteriocinogenic activity was carried out and the effect of rye bran enzymatic hydrolysate (RBEH) on gene expression and bacteriocin production was evaluated. qPCR and RT-PCR was applied for bacteriocin gene detection and their expression quantification. The agar diffusion technique with the test strains of *Bacillus* spp., *Staphylococcus* spp. and *Salmonella enterica* was performed for antimicrobial activity assessment of LAB cultivated in MRS broth and RBEH (processed with proteases and cellulases). The genes of different bacteriocins were revealed for thirteen out of eighteen LAB strains, while the antimicrobial activity was detected only for four of them. The strains of *Lactobacillus paracasei* VKPM B-11657 and *L. salivarius* VKPM B-2214 with unnamed class IIb bacteriocin gene demonstrated the widest spectrum of activity. The growth patterns and bacteriocin gene expression differed between both strains and media. The activity of cell-free supernatants after cultivation in RBEH was slightly lower. However, the test strain of *S. epidermidis* was inhibited by *L. paracasei* cultivated in RBEH but not in MRS. Thus, rye bran can be applied as a sole source of nutrients for LAB fermentation and bacteriocin production.

**Keywords:** lactic acid bacteria; bacteriocins; antimicrobial activity; quantitative polymerase chain reaction; gene expression; green raw materials; rye bran enzymatic hydrolysates

## 1. Introduction

Lactic acid bacteria (LAB) is one of the most important probiotics and is widely used in the food industry due to its bio-preservative ability to inhibit competitive species, including food-borne pathogens as *Listeria* sp., *Clostridium* sp., *Staphylococcus* sp. and *Escherichia* sp. [1]. The preservation effect of LAB in food is determined by the production of antimicrobial substances such as diacetyl, acetoin, organic acids, hydrogen peroxide and bacteriocins. Since LAB are generally regarded as safe (GRAS), their metabolites, including bacteriocins, are the focus of research in food and nutrition, medicine, personal care. Bacteriocins are ribosomally-synthesized bioactive peptides produced by bacteria and display antimicrobial activity against similar or closely related bacterial strains [2]. In addition to antagonistic activity, bacteriocins play an important role as signaling peptides allowing bacteria to control their gene expression in response to their population density through quorum sensing (QS) regulation [3]. Bacteriocins include peptides or proteins of variable molecular weight, biochemical properties, mode of action, spectrum of activity, biosynthesis mechanism and genetic determinants. Biosynthetic clusters of bacteriocins

are located in plasmid or chromosomal DNA and they involve genes encoding bacteriocin precursors (structure gene), post-translational modification enzymes, transport, immune and regulation proteins. Bacteriocin-producing strains are resistant to these antimicrobial peptides due to the presence of immunity proteins on the cell membrane. Bacteriocins are usually divided into three classes, i.e., small post-translationally modified peptides (class I), unmodified peptides (class II) and heat unstable proteins (class III). Unlike traditional antibiotics, bacteriocins are active at nanomolar concentrations, show low toxicity against mammalian cells and can be digested by gastric enzymes [4].

Bacteriocin-producing LAB are considered as an alternative to antibiotics since these strains can improve the gut health, inhibit pathogens, and modulate the immune system [5].

The reduction in manufacturing costs to realize the possibility of utilizing bacteriocin industrial production synthesized by LAB is one of the relevant objectives. Alternative types of raw materials for the cultivation of lactobacilli could be considered as a cost reduction factor. Replacing individual components of standard nutrient media (peptone, yeast extract, meat extract) with alternative sources of nitrogen and vitamins seems prospective. Nevertheless, publications for this issue are limited. The addition of sugar cane pulp hemicellulose hydrolysate to the nutrient medium led to an increase in the expression of PA-1 genes in the *Pediococcus pentosaceous* ET34 strain [6]. The addition of soy extract to MRS broth instead of yeast extract led to an increase in the production of bacteriocin by the *Lactococcus lactis* CCSULAC strain [7]. Fermented barley extract obtained from a barley shochu by-product and its ethanol fractions were evaluated as a medium and supplement, respectively, for nisin A production by *Lactococcus lactis* subsp. *lactis* ATCC 11454 [8]. Cull potatoes enriched with yeast extract, meat or soy peptone, corn steep solid provide enough essential nutrients for the growth of *Lactococcus lactis* subsp. *lactis* ATCC 11454 and nisin biosynthesis stimulation [9]. The growth of six *Lactiplantibacillus plantarum* strains in wheat flour-water extract (compared with standard and modified MRS broth) and the plantaricin activity against a mixture of five *Listeria monocytogenes* strains was studied [10].

In previous research, it has been shown that cereal processing products treated with the enzyme preparations can be used as the only source of nutrients for the cultivation of lactobacilli's various species as well as to obtain their high titer [11], which may also be used for the production of bacteriocin. Nevertheless, obtaining a good composition of nutrient media for the strain growth does not mean a high yield of the target product, especially in the case of such complex molecules as bacteriocins.

The aim of this study was the screening of LAB strains for bacteriocinogenic activity and the evaluation of rye bran enzymatic hydrolysate effect on gene expression and bacteriocin production.

## 2. Materials and Methods

### 2.1. Green Raw Materials and Enzyme Preparations

Rye bran purchased from Productovaya Apteka (Moscow, Russia) was used as green raw material. The composition declared by the manufacturer was as follows (per 100 g): 11.2 g of proteins, 3.2 g of fats, 32 g of carbohydrates. Hydrolysis of rye bran was carried out with Protex 40E enzyme provided by the Russian local office of DuPont—Genencor Danisco (Moscow, Russia) and commercial cellulases Cellic HTec 2 provided by the Russian local office of Novozymes A/S (Moscow, Russia).

### 2.2. Preparation of Enzymatic Bran Hydrolysates

Rye bran was suspended in distilled water in 1:20 ratio. Enzymatic hydrolysis was carried out with stirring (120 rpm) under conditions recommended by the manufacturer. Proteases and cellulases were added consecutively. Hydrolysis with Protex 40E (2.0% of enzyme to the raw material protein) was carried out at pH 8.6 and 60 °C for 90 min. Hydrolysis with Cellic Htec 2 (12.5% of enzyme to the raw material dry matter) was carried out at pH 5.0 and 50 °C for 1.5 h. At the end of hydrolysis, pH was adjusted to 6.8 (using 10% $w/v$ NaOH and 10% $v/v$ $H_2SO_4$) and rye bran enzymatic hydrolysates (RBEH)

autoclaved at 121 °C for 30 min. The pulp was not separated. The concentrations of the soluble nutrient substances were measured. Protein (Lowry) was 5.7 mg/mL and reducing sugars (Fehling) were 22.3 mg/mL.

### 2.3. Bacterial Strains and Growth Conditions

LAB strains were purchased from All-Russian National Collection of Industrial Microorganisms (VKPM, Moscow, Russia). The collection numbers and the sources of origin of the LAB strains are presented in Table 1. The strains were cultured in De Man, Rogosa, Sharpe (MRS) broth [12] at 37 °C for 24 h to obtain the inoculates and for the screening of the genes of bacteriocins. To analyze gene expression and for antimicrobial activity study, lactobacilli were cultured in MRS broth and RBEH medium. The fermentations were carried out in Erlenmeyer flasks at 37 °C without stirring and maintaining pH for 24 h. Sampling for analysis was carried out after 4, 8, 16, 24, 30 h of the cultivation.

**Table 1.** The origins of the organisms used in the study.

| Species | Strains | Origin |
| --- | --- | --- |
| *Lactobacillus acidophilus* | VKPM B-2105 | Fermented dairy product |
| *Lactobacillus acidophilus* | VKPM B-2900 | Human |
| *Lactobacillus acidophilus* | VKPM B-6551 | Human |
| *Lactobacillus acidophilus* | VKPM B-6552 | Human |
| *Lactobacillus buchneri* | VKPM B-7641 | Tomato pulp |
| *Lactobacillus delbrueckii* subsp. *bulgaricus* | VKPM B-11948 | Bulgarian yoghurt |
| *Lactobacillus casei* | VKPM B-2872 | Human |
| *Lactobacillus casei* | VKPM B-2873 | Human |
| *Lactobacillus fermentum* | VKPM B-8724 | Fermented beets |
| *Lactobacillus gallinarum* | VKPM B-10907 | Chicken crop |
| *Lactobacillus graminis* | VKPM B-11233 | Grass silage |
| *Lactobacillus paracasei* | VKPM B-4079 | Fermented sugar beet tops |
| *Lactobacillus paracasei* | VKPM B-11657 | Human |
| *Lactobacillus plantarum* | VKPM B-11007 | Lactobacterin |
| *Lactobacillus rhamnosus* | VKPM B-8238 | Infant |
| *Lactobacillus reuteri* | VKPM B-9448 | Human |
| *Lactobacillus sakei* | VKPM B-8936 | Italian dry-cured sausage |
| *Lactobacillus salivarius* | VKPM B-2214 | Human |

Test cultures for antimicrobial activity assay were purchased from All-Russian National Collection of Industrial Microorganisms (VKPM, Moscow, Russia). Bacterial strains used in this study included *Bacillus megaterium* VKPM B-3750, *Bacillus coagulans* VKPM B-9868, VKPM B-10468, VKPM B-4521, *Bacillus cereus* VKPM B-8076 (correspond to ATCC 9634, applied to efficacy testing), *Salmonella enterica* serovar Typhimurium VKPM B-5300 (applied to biological testing). Test strains of *Staphylococcus aureus* ATCC 43300 (type strain, applied to efficacy testing), *Staphylococcus epidermidis* ATCC 14990 (applied to susceptibility testing and drug discovery) were purchased from the American Type Culture Collections. The test strains were cultured in Erlenmeyer flasks at a temperature of 37 °C without stirring and maintaining pH for 24 h under microaerophilic conditions.

### 2.4. Enumeration of Lactobacilli

Enumeration of LAB viable cells (CFU/mL) was carried out by plating ten-fold dilutions of the samples on MRS agar. Plates are incubated at 37 °C for 48 h. Counting has been performed in triplicate, and the results are expressed as $\log_{10}(\text{CFU/mL}) \pm \text{SD}$.

### 2.5. Molecular Genetic Research Methods

#### 2.5.1. LAB Biomass Separation for PCR Analyses

After fermentation in MRS, the LAB cells were separated by centrifugation at 7500 rpm for 15 min (Centrifuge 5810 R, Eppendorf SE, Hamburg, Germany). A supernatant was

used for antimicrobial activity assessment and cells were washed twice in sterile PBS (0.1 M, pH = 7.2). When cultivated in RBEH, LAB cells were additionally desorbed from solid particles as described by Macfarlane and Macfarlane [13] with some modifications. The samples were centrifuged and washed with sterile PBS. After second separation the 0.001% $w/v$ solution of cetyltrimethylammonium bromide (CTAB, Merck KGaA, Darmstadt, Germany) and particles were subsequently incubated for 30 min at 37 °C with shaking. Then the samples were centrifuged. The top layer of the insoluble substances was carefully removed by pipetting and the second layer of cells were harvested while the bottom layer contained cell-free solid particles.

### 2.5.2. Identification of Bacteriocin-Related Structural Genes

Real-time PCR (qPCR) was used to screen the probiotic strains for the presence of genes encoding known and predicted *Lactobacillus* bacteriocins, as well as general lantibiotic genes. Bacteriocins structural genes sequences were obtained from NCBI GenBank database. Primers for chromosome or plasmid encoded bacteriocins of lactic acid bacteria were designed using NCBI Primer-BLAST tool (Table 2). Degenerate primers were used to detect the lantibiotic genes. Genomic DNA was isolated using diaGene bacterial DNA extraction kit (Dia-M, Russia) according to the manufacturer's instructions. PCR amplification was performed on CFX96 Touch Deep Well Real-Time PCR Detection System (Bio-Rad, USA) in 25 µL reaction mixtures. Each PCR mix contained 5 µL qPCRmix-HS SYBR 5x (Evrogen, Russia; consists of *Taq* DNA Polymerase, dNTPs, buffer, $MgCl_2$ and SYBR Green I), 1 µL of each primer with final concentration of 200 nM and 1 µL of the cell lysate. PCR conditions were as follows: initial denaturation for 3 min at 95 °C, followed by 36 cycles: denaturation at 95 °C for 15 s, annealing at 55 °C for 15 s and elongation at 72 °C for 30 s. In the case of degenerate primers, the final concentration of 100 nM was used and PCR condition were as follows: initial denaturation for 3 min at 95 °C, followed by 36 cycles: denaturation at 95 °C for 30 s, annealing at 40 °C for 30 s and elongation at 65 °C for 30 s. The specificity of amplification was verified by melting curve analysis from 50 °C to 90 °C at 0.2 °C/s following the qPCR and 2.5% agarose gel electrophoresis.

### 2.5.3. Extraction of Total RNA

Total bacterial RNA isolation included mechanical lysis using bead beating and column-based extraction of RNA from cell lysate. Two mL aliquots of bacterial cells that had been stored at −20 °C were centrifuged during 10 min at 10,000 rpm, resuspended in 2 mL sterile saline solution and recentrifuged in the same condition. Washed cells were resuspended in 650 µL of lysis buffer (Dia-M, Moscow, Russia) and all aliquots were transferred to Microbial Lysis Tubes from AllPrep PowerFecal DNA/RNA Kit (QIAGEN GmbH, Hilden, Germany) with beating beads and were bead beaten in a horizontal position in a vortex mixer at 3000 rpm for 15 min. The lysates were centrifuged for 2 min at 20,000× $g$ and aliquots of supernatant were used for nucleic acids extraction. RNA was purified using diaGene RNA extraction kit from cell cultures (Dia-M, Moscow, Russia) according to the manufacturer's instructions. All nucleic acid extracts were treated with 2 units of DNase I (Thermo Fisher Scientific Baltics UAB, Vilnius, Lithuania) at 37 °C for 30 min in order to remove contaminating DNA, then DNase was inactivated by adding 50 mM EDTA at 65 °C for 10 min. Purified RNA was analyzed for quantity and quality with NanoPhotometer N60-Touch (Implen GmbH, München, Germany) at 260/280 and 260/230 absorbance ratios. The integrity of isolated RNA samples as well as their DNA contamination was tested by using 2% agarose gel electrophoresis. The gels were stained with ethidium bromide (0.15 µg/mL) and then visualized by UV-transillumination.

Table 2. Primers used in bacteriocin structure genes screening.

| No | Product | Gene | Primer Sequence (5′-3′) | Size, bp | Reference |
|---|---|---|---|---|---|
| 1. | Acidocin 8912 | *acdT* | F: AAGAATTAGCATTAATTTCTGGGGG<br>R: CAGTATAACGAAGGCTTTCCCA | 96 | This study |
| 2. | Acidocin B | *acdB* | F: GCCACAGCGAACATTTATTGGA<br>R: ACGCCCAAAATAGCAGCAAAG | 131 | This study |
| 3. | Bacteriocin BacSJ2-8 | *bacSJ2-8* | F: ACTACCCCAGCCATCTCCAA<br>R: TGGAGAGACAAGAGGGGTCA | 90 | This study |
| 4. | Gassericin T | *gasT* | F: GGAGTAGGTGGAGCGACAGT<br>R: TCCACCAGTAGCTGCCGTTA | 126 | This study |
| 5. | Helveticin J | *hlvJ* | F: TCAAACAAACCAAAGTGACC<br>R: CAAGTTGGTGCAGTAAATGGTG | 440 | This study |
| 6. | Helveticin homolog | *Hlv* | F: CCATTTGTTCGCCATACCAGCA<br>R: CCACACCATCATTCAGCCGTTC | 198 | This study |
| 7. | Lactococcin 972 family | LacC0470_08940 | F: GCTTATACAGTTGATGTGCAAGGT<br>R: CTTGATGCGCTGTATGCTCC | 128 | This study |
| 8. | Lactococcin 972 family | LG542_08235 | F: TTTCTTACACCAGCACCCCATC<br>R: ATTCGGGATTCAGTGGAACAAAG | 100 | This study |
| 9. | PapA | *papA* | F: CTTGTGGCAAACATTCCTGCT<br>R: CTTGATGTCCACCAGTAGCCC | 95 | This study |
| 10. | Plantaricin A | *plnA* | F: AGCAACTTAGTAATAAGGAAATGCAAA<br>R: ACAGTTTCTTTACCTGTTTAATTGCAG | 102 | [14] |
| 11. | Plantaricin E | *plnE* | F: ATACCACGAATGCCTGCAAC<br>R: ATCTGGTGGTTTTAATCGGGG | 93 | This study |
| 12. | Plantaricin F | *plnF* | F: CTAATGACCCAATCGGCAGG<br>R: ATGCTATTTCAGGTGGCGTTTT | 94 | This study |
| 13. | Plantaricin J | *plnJ* | F: ATAATAAGTTGAACGGGGTTGTTGG<br>R: TGCCAGCTTCGCCATCATAAA | 90 | This study |
| 14. | Plantaricin K | *plnK* | F: TAATCCCTTGAACCACCAAGCA<br>R: TAACTGCTGACGCTGAAAAGAA | 124 | This study |
| 15. | Sakacin A | *sakA* | F: TTCCAGCTAAACCACTAGCCC<br>R: AAATGTTGGGTAAATCGGGGTG | 82 | This study |
| 16. | Sakacin P | *sppA* | F: AACAGCAATTACAGGTGGAAAA<br>R: TATTTATTCCAGCCAGCGTTTC | 150 | This study |
| 17. | Sakacin T alpha | *sakTα* | F: AGAAGAATTGGTACTTGTAGTCGGT<br>R: CCTGCTCCTGTACCAGCAATAC | 90 | This study |
| 18. | Sakacin T beta | *sakTβ* | F: TAATTGGGGATCAGTCGTGGG<br>R: GTCCTGCACCGACTAAGCATC | 104 | This study |
| 19. | Sakacin Q | *sakQ* | F: TTGGTAAATGTGTAGTTGGTGCTTG<br>R: CCATTCCCCAGAGACCACCA | 81 | This study |
| 20. | Sakacin X | *sakX* | F: TTGTCGGGGGAAAATACTACGG<br>R: CAGCTCCACCGGTAGTCAAA | 126 | This study |
| 21. | Salivaricin ABP-118 α | *abp118α* | F: GCAAAGGTTGATGGTGGGAAAC<br>R: TAAGTGCTCCGCCTACCATTC | 145 | This study |
| 22. | class II b bacteriocin | LRHK_2402 | F: GATCGTTCCCATCCTTGCTTG<br>R: ATCCCTGTCGCTGCTATTCTTG | 95 | This study |
| 23. | class II b bacteriocin | LCA12A_2788 | F: AATTAGTGCTGAGACACAAGGA<br>R: GTCTTAATCCAAGGAGGAGCCA | 154 | This study |
| 24. | Class I-Type I Lantibiotic | *lanB* | F: TATGATCGAGAARYAKAWAGATATGG<br>R: TTATTAIRCAIATGIAYDAWACT | ~250 | [15] |
| 25. | Class I-Type I Lantibiotic | *lanC* | F: TAATTTAGGATWISYIMAYGG<br>R: ACCWGKIIIICCRTRRCACCA | ~400 | [15] |
| 26. | Class I-Type II Lantibiotic | *lanM* | F: ATGCWAGWYWTGCWCATGG<br>R: CCTAATGAACCRTRRYAYCA | 330 | [15] |

### 2.5.4. Quantitative Reverse Transcription Quantitative PCR and Gene Expression Analysis

Two housekeeping genes (*ef-Tu* and 16S rRNA) were selected for bacteriocin gene expression analysis (Table 3). RNA detection was performed using one-step RT-qPCR kit in 25 μL reaction mixture. Each PCR mix contained 6 μL OneTube RT-PCRmix (Evrogen,

Moscow, Russia) (consists of MMLV Reverse Transcriptase, *Taq* DNA Polymerase, dNTPs, buffer, MgCl$_2$), 0.25 μL of 50X SYBR Green I (Evrogen, Moscow, Russia), 2 μL of each primer with final concentration of 400 nM and 1 μL of RNA solution (100–200 ng). The thermal cycling conditions were as follows: 42 °C for 15 min (reverse transcription), 1 min at 95 °C (*Taq* polymerase activation and reverse transcriptase inactivation), followed by 45 cycles of 15 s at 95 °C (denaturation), 15 s at 55 °C (annealing) and 20 s at 72 °C (elongation). The specificity of amplification was verified by melting curve analysis from 58 °C to 95 °C. The data were analyzed using Bio-Rad CFX Manager 3.1 Software. Relative quantities of bacteriocin-related genes were calculated using delta-Ct method while accounting for differences in primer efficiencies (RQ = E$^{ΔCt}$). The values were normalized to housekeeping genes quantities to count the relative expression of bacteriocin genes. The efficiency is calculated based on serial dilutions of a known amount of RNA template (E = 10$^{-1/slope}$).

**Table 3.** Primers used in bacteriocin genes expression analysis.

| Function | Product | Gene | Primer Sequence (5′-3′) | Size, bp | Reference |
|---|---|---|---|---|---|
| Housekeeping genes | 16S rRNA | *16S rRNA* | F: ACTCCTACGGGAGGCAGCAG<br>R: GTATTACCGCGGCTGCTGG | 200 | [16] |
| | elongation factor Tu | *Ef-Tu* | F: TCGAYGCTGCWCCDGAAGA<br>R: TGGCATWGGRCCATCAGTWG | 194 | [17] |
| Bacteriocin-related gene | class II b bacteriocin | LCA12A_2788 | F: ATTAGTGCTGAGACACAAGGA<br>R: TCTTAATCCAAGGAGGAGCCA | 154 | This study |

*2.6. Antimicrobial Activity Assay*

The antimicrobial activity of LAB cell-free supernatant against test strains (bacteriocinogenic activity) was assessed by the agar diffusion method according to Schillinger & Lücke [18] with some modifications. One mL of a night testing culture (approximately of 4·10$^6$ CFU/mL) was inoculated in 19 mL of melted (about 40 °C) LB-agar (*Bacillus* spp. strains), mannitol salt agar (*Staphylococcus* spp. strains) or Endo agar (for *Salmonella*), stirred, plated into sterile Petri dish and kept at room temperature for about 20 min to solidify. To obtain a cell-free supernatant (CFS), the culture fluid of lactobacilli was centrifuged at 7500 rpm for 15 min (Centrifuge 5810 R, Eppendorf SE, Hamburg, Germany). The supernatants were neutralized with 5 M NaOH solution to pH 6.5 and were sterilized by filtration (0.2 μm, Spartan 13, Whatman GmbH, Dassel, Germany). Wells (6 mm) were cut in solidified agar using sterile 1 mL cut-off pipette tips, filled with 55 μL of the neutralized supernatant. Metronidazole 5 mg/mL solution (PJSC Sintez, Kurgan, Russia) and 10 mg/mL solution of Ciprofloxacin (HPLC grade, Merk, Saint Louis, MO, USA) were used as a control samples. The Petri dishes were kept at room temperature for 2 h to ensure diffusion. They were incubated under aerobic conditions at 37 °C for 24 h and the inhibition zones (no growth of the test strain) diameters were measured. Antimicrobial activity of the samples of sterilized RBEH (blank control) was assessed with the same technique.

**3. Results**

*3.1. Screening of Lactic Acid Bacteria Strains for Bacteriocin Genes*

The LAB strains were tested for the presence of genes encoding known *Lactobacillus* bacteriocins (*acdT, acdB, bacSJ2-8, gasT, hlvJ, hlv, papA, plnA, plnE, plnF, plnJ, plnK, sakA, sakQ*) and predicted bacteriocins lactococcin 972 family and class II b bacteriocins, as well as lantibiotic-related genes *lanB, lanC* (class I type I lantibiotics) and *lanM* (class I type II lantibiotics). The results are presented in Table S1.

Plantaricin-related genes *plnA, plnE, plnF, plnJ* were found in *Lactobacillus plantarum* B-11007. Plantaricin EF *(plnEF)* and plantaricin JK *(plnJK)* are both two-peptide bacteriocins that act as synergetic peptides [19]. The production of these bacteriocins induces peptide plantaricin A *(plnA)* [20].

Positive results for the presence of plasmid encoded bacteriocin BacSJ and bacteriocin of lactococcin 972 family was obtained for *Lactobacillus graminis* B-11233. According to the literature data, BacSJ is a bacteriocin produced by *Lactobacillus paracasei* subsp. *paracasei* [21]. It is encoded by the *bacSJ2-8* gene on the pSJ2–8 plasmids [22]. The *bacSJ2-8/bacSJ2-8i* gene cluster was present in 49 (94.23%) of the 52 tested strains [23]. Lactococcin 972 is a plasmid-encoded non-lantibiotic bacteriocin produced mainly by *Lactococcus* sp. Predicted gene of two-peptide bacteriocin (class IIb) specific to *Lactobacillus paracasei* was found in *Lactobacillus gallinarum* B-10907, *Lactobacillus paracasei* B-4079, *L. paracasei* B-11657 and *Lactobacillus salivarius* B-2214. A similar predicted gene specific to *Lactobacillus rhamnosus* were detected in *Lactobacillus rhamnosus* B-8238. These potential gene products contain GxxxG and AxxxA motifs, that allow contact between the two helices in two-peptide bacteriocins [1].

*Lactobacillus fermentum* B-8724 was positive tested for *hlvJ* gene encoding helveticin J, non-lytic heat-sensitive protein. Previously, the gene *hlvJ* was detected in *Lactobacillus acidophilus* B-2105, *L. acidophilus* B-2900, *L. acidophilus* B-6551 and *L. acidophilus* B-6552. The *gasT* gene encoding gassericin T, a putative two-peptide bacteriocin, was found in *Lactobacillus buchneri* B-7641. The lantibiotic-associated genes were not detected in the strains.

### 3.2. Screening Lactic Acid Bacteria Strains for Bacteriocinogenic Activity

The ability of lactobacilli to produce bacteriocins (bacteriocinogenic activity) can be assessed as antimicrobial activity of their CFS (adjusted to neutral or slightly acidic pH) against test strains by agar diffusion method [18]. The test strains of different groups were used. Gram-negative *S. enterica* can cause gastroenteritis, vomiting, and diarrhea, is often antibiotic resistant, and forms a biofilm [24]. The remaining strains are Gram-positive. Common food spore forming contaminant *B. cereus* produces two types of toxins that cause vomiting and diarrhea types intoxication [25]. *B. coagulans* and *B. megaterium* are not usually pathogenic but often cause food spoilage [26]. Both *S. aureus* and coagulase-negative *Staphylococcus* spp. can cause nosocomial infections [27].

The solution (5 mg/mL) of metronidazole was considered as a control for *Bacillus* spp. and *Staphylococcus* spp. strains. The antimicrobial activity of metronidazole against diarrhea causing clinical isolates of *B. cereus* and *S. aureus* was observed [28]. The solution (10 mg/mL) of Ciprofloxacin (control) was applied in the tests with *Salmonella* [29]. The diameters of growth inhibition zones for the CFS of lactobacilli cultivated in MRS broth were assessed. The antibacterial activity was revealed for the CFS of *L. acidophilus* VKPM B-2105, *L. paracasei* VKPM B-11657, *L. salivarius* VKPM B-2214 and *L. plantarum* VKPM B-11007 (Table 4). Antagonistic activity of the CFS of other studied lactobacilli against the test strains was not detected. None of the CFS inhibited growth of *B. cereus* and *S. aureus*. The CFS of *L. acidophilus* VKPM B-2105 had highest antimicrobial activity against *S. epidermidis*. The CFS of *L. paracasei* VKPM B-11657 and *L. salivarius* VKPM B-2214 showed activity against the wide spectrum of the test strains of *Bacillus*. The test strain of *S. enterica* VKPM B-5300 was strongest suppressed by the CFS of *L. salivarius* VKPM B-2214.

Based on the results of antimicrobial analysis on agar plate and genetic screening, *Lactobacillus paracasei* B-11657 and *Lactobacillus salivarius* B-2214 strains were selected for the next stage of the study.

**Table 4.** Diameters of growth inhibition zones for the CFS of lactobacilli cultivated in MRS against test strains.

| Test Strains | Diameter of Growth Inhibition Zone ± SD (mm) | | | | |
|---|---|---|---|---|---|
| | *L. acidophilus* VKPM B-2105 | *L. paracasei* VKPM B-11657 | *L. salivarius* VKPM B-2214 | *L. plantarum* VKPM B-11007 | Control * |
| *B. megaterium* VKPM B-3750 | 7.3 ± 0.6 | 13.8 ± 0.5 | 12.2 ± 0.5 | 7.2 ± 0.5 | 21.0 ± 0.9 |
| *B. coagulans* VKPM B-4521 | - *** | 14.0 ± 0.7 | 15.7 ± 0.6 | - | 12.0 ± 0.7 |
| *B. coagulans* VKPM B-9868 | 14.0 ± 1.0 | 11.7 ± 0.9 | 9.8 ± 0.9 | 7.5 ± 0.6 | 7.3 ± 0.5 |
| *B. coagulans* VKPM B-10468 | - | 7.2 ± 0.5 | 8.0 ± 0.7 | - | 11.8 ± 0.9 |
| *B. cereus* ATCC 9634 | - | - | - | - | 11.8 ± 0.9 |
| *S. aureus* ATCC 4330 | - | - | - | - | 10.7 ± 0.6 |
| *S. epidermidis* ATCC 12228 | 16.2 ± 0.5 | - | 7.2 ± 0.9 | - | 19.5 ± 1.2 |
| *S. enterica* VKPM B-5300 ** | 7.8 ± 0.9 | 7.5 ± 0.6 | 12.2 ± 0.5 | 7.3 ± 0.6 | 20.5 ± 1.1 |

* Metronidazole solution (5 mg/mL); ** ciprofloxacin solution (10 mg/mL) was used as a control; *** antimicrobial activity is not detected.

### 3.3. Dynamics of L. salivarius VKPM B-2214 and L. paracasei VKPM B-11657 Growth and Bacteriocin Genes Expression during Cultivation in MRS Broth and Rye Bran Hydrolysate

The high manufacturing costs of bacteriocins are one of the problems in their production in the industrial scale. Rye bran is a by-product of grain processing. The bacteriocinogenic activity of lactobacilli in RBEH compared with MRS was studied to assess the prospects of expensive commercial nutrient media replacing and reducing the manufacturing costs. The preliminary study on the liquid phase of RBEH fermented with lactobacilli was carried out and growth inhibition zone was not detected for all test strains. Therefore, the solid of RBEH was not separated further (submerged heterogeneous fermentation).

The growth patterns of *L. paracasei* VKPM B-11657 and *L. salivarius* VKPM B-2214 varied for different strains and media (Figure 1a,b). The stationary phase of *L. paracasei* in RBEH was observed at 8 h of fermentation and at 16 h in other cases. The cell count in the stationary phase was greater in MRS compared to RBEH. Cell lysis was revealed after 24 h for *L. salivarius* in MRS and *L. paracasei* in RBEH.

The expression of gene encoding two-peptide bacteriocin was analyzed by measuring the mRNA in *Lactobacillus* strains (Figure 1c,d). In order to account for the difference of total RNA amount in the samples, expression level of bacteriocin gene was normalized to the expression 'housekeeping' gene (stably expressed gene). It was noticed that transcription level of *ef-Tu* gene encoding elongation factor Tu was close to transcription level of bacteriocin gene in some cases, so only 16S rRNA gene was used as a housekeeping gene in the calculations of normalized expression.

To confirm the effect of growth media on antimicrobial activity of lactobacilli, expression level of bacteriocin gene was measured in cells cultured both in MRS broth and RBEH. Maximum transcription level of bacteriocin gene was at 8 h in both strains grown in media containing hydrolysates and it decreased evenly up to 30 h. Expression of the same gene in lactobacilli cells grown in MRS broth was high at 8 h for *L. paracasei* B-11657 and at 8 h and 24 h for *L. salivarius* B-2214, while marked level of expression was maintained up to 30 h. In the case of *L. paracasei* B-11657, transcription level of bacteriocin gene at 8 h in RBEH was greatly higher than in MRS.

No inhibition zones of the test strains were detected for CFS at 4 and 8 h of cultivation and for CFS of LAB cultivated in RBEH at 30 h. The activity (Table 5) of CFS after LAB cultivation in RBEH against the test strains of *Bacillus* spp. was mostly comparable to MRS at 16 h, but less at 24 h. Moreover, the inhibition of *S. enterica* by CFS of *L. paracasei* VKPM B-11657 was detected at 16 h and 24 h but not at 30 h in MRS, and only at 16 h in RBEH. Contrarily, the strain of *L. paracasei* VKPM B-11657 inhibited the growth of *S. epidermidis* when fermentation was carried out in RBEH but not in MRS. While the maximum expression level in RBEH was observed at 8 h with a subsequent decrease, the activity was first detected at 16 h, reached a maximum at 24 h and then disappeared. In

contrast, in MRS, expression was observed in all samples and changed less dramatically, though the peaks of activity were revealed at 24 h. In general, correlation between the level of bacteriocin gene expression and antimicrobial activity cannot be recognized.

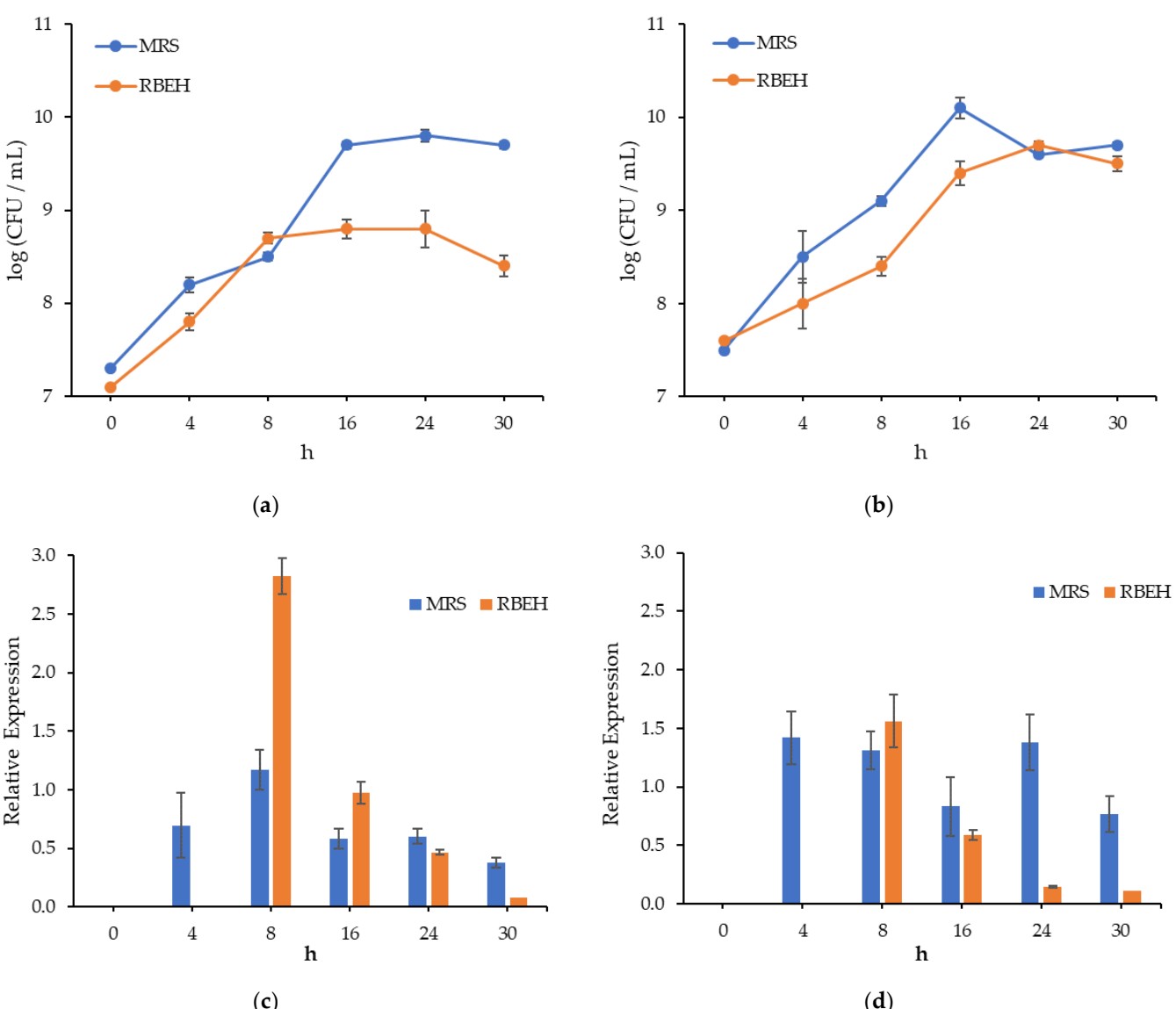

**Figure 1.** Growth dynamics of *Lactobacillus paracasei* B-11657 (**a**) and *Lactobacillus salivarius* B-2214 (**b**) and relative expression of bacteriocin gene in *Lactobacillus paracasei* B-11657 (**c**) and in *Lactobacillus salivarius* B-2214 (**d**) normalized to the expression of 16S rRNA gene during cultivation in MRS broth and RBEH.

**Table 5.** Diameters of the test strains growth inhibition zones for the CFS of lactobacilli cultivated in MRS and RBEH.

| Test Strains | Diameter of Growth Inhibition Zone ± SD (mm) | | | | | | | | | | RBEH |
| --- | --- | --- | --- | --- | --- | --- | --- | --- | --- | --- | --- |
| | *L. paracasei* VKPM B-11657 | | | | | *L. salivarius* VKPM B-2214 | | | | | before |
| | MRS | | | RBEH | | MRS | | | RBEH | | |
| | 16 h | 24 h | 30 h | 16 h | 24 h | 16 h | 24 h | 30 h | 16 h | 24 h | Inoculation |
| *B. megaterium* VKPMB-3750 | 11.7 ± 1.3 | 18.8 ± 1.0 | 18.0 ± 0.9 | 11.7 ± 0.9 | 8.7 ± 0.9 | 11.0 ± 0.9 | 20.8 ± 1.0 | 19.2 ± 0.8 | 7.5 ± 0.7 | 9.3 ± 0.5 | - |
| *B. coagulans* VKPM B-4521 | 12.8 ± 0.8 | 18.0 ± 0.7 | 17.2 ± 0.8 | 12.7 ± 0.5 | 11.2 ± 0.8 | 16.3 ± 1.1 | 25.0 ± 0.7 | 23.0 ± 0.7 | 11.7 ± 0.9 | 13.2 ± 0.8 | - |
| *B. coagulans* VKPM B-9868 | 10.0 ± 1.3 | 24.7 ± 0.9 | 24.0 ± 0.9 | 12.3 ± 0.9 | 9.7 ± 1.1 | 9.2 ± 0.8 | 27.0 ± 0.9 | 24.3 ± 0.9 | 10.7 ± 0.9 | 12.0 ± 0.7 | - |
| *B. coagulans* VKPM B-10468 | 7.5 ± 0.6 | 11.0 ± 0.9 | 9.7 ± 0.5 | 12.0 ± 0.9 | 10.0 ± 0.7 | 7.2 ± 0.4 | 11.0 ± 0.4 | 10.2 ± 0.4 | 11.5 ± 1.1 | 14.2 ± 0.8 | - |
| *B. cereus* ATCC 9634 | - * | - | - | - | - | - | - | - | - | - | - |
| *S. aureus* ATCC 4330 | - | - | - | - | - | - | - | - | - | - | - |
| *S. epidermidis* ATCC 14990 | - | - | - | 13.8 ± 1.0 | 13.3 ± 0.9 | 8.0 ± 0.7 | 11.8 ± 0.4 | 9.8 ± 0.8 | 8.7 ± 0.5 | 15.2 ± 0.8 | - |
| *S. enterica* VKPM B-5300 | 7.8 ± 0.8 | 10.2 ± 0.8 | - | 10.7 ± 0.9 | - | - | - | - | - | - | - |

* Antimicrobial activity is not detected.

## 4. Discussion

The PCR screening of the examined strains of LAB from VKPM for the presence of genes encoding known and assumed bacteriocins produced by *Lactobacillus* spp., as well as genes associated with lantibiotics, showed that many of them have the potential ability to produce bacteriocins. The most common helveticin genes were detected in the strains of *L. acidophilus* and *L. fermentum*, while unnamed class II b bacteriocins genes were detected in the strains of *L. salivarius*, *L. paracasei*, *L. gallinarum* and *L. rhamnosus*. Plantaricin family genes were discovered in *Lactobacillus plantarum* B-11007.

Bacteriocins of various lactobacilli differ from each other in their unique genetic, structural, biochemical, metabolic and ecological activity [30]. Antimicrobial activity of LAB is characterized by high selectivity and specificity in relation to a rather limited spectrum of microorganisms. In many cases, sensitivity is strictly strain-specific. This can be used against opportunistic and pathogenic microorganisms, including bacteria with multiple antibiotic resistance [31].

The strains of *Lactobacillus paracasei* VKPM B-11657 and *Lactobacillus salivarius* VKPM B-2214 showed the best results against the wide spectrum of the test strains. The strains of *Bacillus coagulans* that can cause food spoilage were recognized as nisin sensitive [32], meanwhile bacteriocin enterocin AS-48 was only effective against vegetative cells, not the spores [33]. The class IIb bacteriocin produced by *Lactiplantibacillus plantarum* RUB1 was found to inhibit the growth of the test strains of *Staphylococcus aureus* ATCC 6538, *Bacillus cereus* ATCC 14579 and *Salmonella* serotype Typhimurium ATCC 14028 [34].

It is worth mentioning that the bacteriostatic activity of bacteriocins depends on many factors, including not only the dose and degree of bacteriocins purification, but as well as the stage of the test cultures growth, temperature and pH (medium acidity) [31]. Currently, the industrial bacteriocins production by microbial fermentation is still limited, which is explained by the low yield and high manufacturing costs [35]. The bacteriocins formation by LAB depends on the total bacterial biomass. This usually corresponds to primary metabolites, which biosynthesis occurs during the exponential growth phase and finished after reaching the stationary phase. However, high cell yield does not necessarily lead to high antimicrobial activity [36]. Most class II bacteriocins are regulated by a quorum sensing (QS) system whose initiation can be induced by environmental factors and other bacterial strains [36]. A lot of research is dedicated to the influence of temperature, pH and properties of the nutrient medium (which is considered as a key factor) [35]. Additionally, within the framework of such studies, the possibility and potential of using plant raw materials as

the main component of nutrient media for the bacteriocinogenic strains cultivation and the production of bacteriocins is being studied in order to reduce production expenses [10].

A comparative assessment effect of the standard MRS broth and RBEH medium on the growth and antimicrobial activity of *L. paracasei* VKPM B-11657 and *L. salivarius* VKPM B-2214 showed that both values were higher in MRS. The production of bacteriocins by *Lactobacillus* cultures may depend on many factors. Their production can be stimulated by the presence of various amino acids in the medium, for example, glycine and histidine [37] and inorganic ions [38].

The expression of the corresponding bacteriocin genes should better predict the expression of antimicrobial activity. However, even in this case, simultaneous correlations of expression and activity may not be observed, which is explained by many factors [39].

In this research, when comparing the expression of the bacteriocin gene and antimicrobial activity, there was a discrepancy in timing between the maximum level of expression and the maximum of antimicrobial activity. In particular, the maximum expression of the class IIb bacteriocin gene was observed at 8 h of cultivation in RBEH, and antimicrobial activity was often maximum at 16 h for *L. paracasei* VKPM B-11657 or 24 h for *L. salivarius* VKPM B-2214.

Timing difference between bacteriocin genes expression and antimicrobial activity observed was also revealed in other studies. The expression of genes responsible for the synthesis of bacteriocin gassericin K7 by the *Lactobacillus gasseri* K7 strain was measured and the maximum expression of gassericin K7 genes involved in production, transport and immunity was observed after 4 h of cultivation (in MRS), while the maximum antimicrobial activity was detected after 8 h [40]. It is necessary to take into consideration the mechanism of bacteriocin synthesis, its post-translational modifications and the mechanism of secretion, as well as the physico-chemical conditions and the presence of inhibitory substances in the medium [41]. Class IIb bacteriocins are synthesized from two precursor peptides encoded by two different genes, then transferred across the membrane with simultaneous cleaving of the leader peptide by means of a carrier protein (ABC-transporter). Moreover, in some two peptide bacteriocins, such as plantaricin EF and plantaricin JK, the gene encoding the transport protein is located in an operon separate from the structural genes [20]. For a more complete analysis of the bacteriocin genes expression, it is reasonable to focus on the transcription level of other the biosynthetic cluster genes, including transport, regulatory and immunity proteins. The probability of the presence in the strains of other antimicrobial substances not detected in this study also should be considered. These substances can be produced at a later stage of population growth. This may be the subject of further study.

In our study, correlation between the level of bacteriocin gene expression and antimicrobial activity cannot be recognized. In the research [10], the effect of substrate, temperature, initial pH value, concentration of NaCl and *Lactiplantibacillus plantarum* strains on plantaricin activity was evaluated in combination with the analysis of the transcriptomic response encoding their *pln* genes after 21 h of growth. A strong correlation was observed between the transcription of genes belonging to the same locus, but the correlation between the activity of plantaricin and the transcription of the considered genes was weak. The inability to correlate bacteriocin activity with gene expression was also stated by Hurtado et al. [42] and Paramithiotis et al. in the study of radish (*Raphanus sativus*) roots fermentation [43].

## 5. Conclusions

A study on eighteen strains of LAB of different origin showed that thirteen of them have genes of different bacteriocins. Nevertheless, bacteriocinogenic activity against *Bacillus* spp., *Staphylococcus* spp. and *Salmonella enterica* was detected only for four strains when cultured in a standard MRS medium. The strains of *Lactobacillus paracasei* VKPM B-11657 и *Lactobacillus salivarius* VKPM B-2214 with class IIb bacteriocin gene demonstrated the widest spectrum of antimicrobial activity. The growth patterns and bacteriocin gene expression levels differed between both strain and MRS or RBEH fermentation media. Nev-

ertheless, a delay in the manifestation of antimicrobial activity was observed in comparison with expression in all variants. The activity of cell-free supernatants after cultivation in RBEH was slightly lower than in MRS. However, the strain of *L. paracasei* VKPM B-11657 inhibited the growth of the test strain of *S. epidermidis*, when fermentation was carried out in RBEH but not in MRS. Thus, rye bran is of interest as a possible sole source of nutrients for the fermentation of LAB and the production of bacteriocins. Further studies can be devoted to the influence of culture conditions on the expression of the responsibility for assembly and transport of antimicrobial peptide, as well as to explain the effect of the solid during fermentation.

**Supplementary Materials:** The following supporting information can be downloaded at: https://www.mdpi.com/article/10.3390/fermentation8120752/s1, Table S1: screening of lactic acid bacteria strains for known and predicted bacteriocin genes.

**Author Contributions:** Conceptualization, N.Y.K. and B.A.K.; methodology, A.V.B. and N.Y.K.; validation, A.V.B. and N.Y.K.; formal analysis, Y.M.A.; investigation, J.M.E., M.V.R., M.A.C. and M.A.K.; resources, Y.M.A.; data curation, I.V.S. and A.V.B.; writing—original draft preparation, N.Y.K. and M.V.R.; writing—review and editing, I.V.S. and V.I.P.; visualization, A.V.B. and B.A.K.; supervision, V.I.P.; project administration, B.A.K.; funding acquisition, V.I.P. All authors have read and agreed to the published version of the manuscript.

**Funding:** The research was sponsored by Russian Science Foundation (Project No. 21-19-00367).

**Institutional Review Board Statement:** Not applicable.

**Informed Consent Statement:** Not applicable.

**Data Availability Statement:** Data is contained within the article and the Supplementary Materials.

**Conflicts of Interest:** The authors declare no conflict of interest.

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
