# Peer review of "Evaluation of Rye Bran Enzymatic Hydrolysate Effect on Gene Expression and Bacteriocinogenic Activity of Lactic Acid Bacteria"

_fermentation, doi:10.3390/fermentation8120752_

Round 1
Author Response
Dear reviewer,
Thank you for the interest in our manuscript. Your comments and suggestions are very important for us. We tried to give as complete and detailed an answer as possible and improve the manuscript in accordance with the recommendations. Please see all responses in the attachment and in the revised manuscript.

Reviewer 2 Report
The manuscript submitted by Julia M. Epishkina et al. describes the potential use of an economical culture medium for bacteriocin production. Initial seleccion of strains was performed on the characterisation of bacteriocin-encoding genes and their phenotypic expression.
Bacteriocin production in the BHM medium and MRS medium (commonly used in LAB production) was compared. The ms is well conducted. The presentation is correct, as well as its conclusions.
Author Response
Dear reviewer,
Great thanks for high appreciation our study and the MS.
Round 2
Reviewer 1 Report
This manuscript is suggested to be accepted in present form.